# Robust Symbol Timing Synchronization for Initial Access under LEO Satellite Channel

**DOI:** 10.3390/s23198320

**Published:** 2023-10-08

**Authors:** Pansoo Kim, Hyuncheol Park

**Affiliations:** 1Department of Information and Communications Engineering, Korea Advanced Institute of Science and Technology (KAIST), Daejeon 34141, Republic of Korea; 2Satellite Communication Research Division, Electronics and Telecommunications Research Institute (ETRI), Daejeon 34129, Republic of Korea; 3Department of Electrical Engineering, Korea Advanced Institute of Science and Technology (KAIST), Daejeon 34141, Republic of Korea

**Keywords:** DVB-RCS2, LEO satellite, demodulator, Doppler, symbol timing recovery (STR), Non-Terrestrial Network (NTN), *FFT* (Fast Fourier Transform)

## Abstract

This paper proposes a robust symbol timing synchronization scheme for return link initial access based on the Digital Video Broadcasting-Return Channel via Satellite 2nd generation (DVB-RCS2) system for the Low Earth Orbit (LEO) satellite channel. In most cases, the feedforward estimator structure is considered for implementing Time Division Multiple Access (TDMA) packet demodulators such as the DVB-RCS2 system. More specifically, the Non-Data-Aided (NDA) approach, without using any kind of preamble, pilot, and postamble symbols, is applicable for fine symbol timing synchronization. However, it hinders the improvement in estimation accuracy, especially when dealing with short packet lengths during the initial access from the User Terminal (UT) to the Gateway (GW). Moreover, when a UT sends a short random access packet for initial access or resource request to the LEO satellite channel, the conventional schemes suffer from a large Doppler error depending on UT’s location in a beam and satellite velocity. To ameliorate these problems, we propose a novel symbol timing synchronization algorithm for GW, and its advantage is confirmed through computer simulation.

## 1. Introduction

In the last decades, many private companies and public organizations have provided connectivity services based on satellite communication networks. In recent years, LEO satellite networks based on large multi-spot beams using Ku and Ka bands have been deployed or are being developed and will be operational over the coming years. The representative companies are OneWeb, Starlink of Space-X, Kuiper of Amazon, and Lightspeed of Telesat LEO [1]. They can offer higher data rates in the range of more than 150 Mbps per user and lower latency in a few tens of milliseconds. In light of improved data rates and reduced latency compared to conventional Geostationary Earth Orbit (GEO) satellite communication systems, recent research efforts have vigorously explored the integration of satellite–terrestrial networks [2,3,4,5]. This endeavor aims to extend service coverage to underserved areas where terrestrial networks are not deployed. Meanwhile, according to the available literature, all other companies except for Telesat LEO have developed communication networks and user terminals utilizing proprietary specifications rather than adhering to commercial standard technologies such as 3GPP and DVB. In the context of the 3GPP NTN standard, it is noteworthy that the development of the first RAN standard to facilitate LEO satellite communication was initiated in release 17 and is currently in progress in the stage of release 18 [6]. Regarding the 3GPP NTN standard activities, the specification works have been undertaken to support the compatibility through software updates without modification of the existing hardware of TN’s UEs. Typically, network timing and frequency synchronization are achieved with the aid of the assumption that satellite UEs of handheld or VSAT type are equipped with GNSS receivers. However, in the context of the TDMA-based DVB-S2x/RCS2 standard, which has been mainly used for GEO satellite communication, network synchronization can be achieved by means of periodic broadcasting of internal highly accurate clock information from the gNB and proper acquisition from the UE without relying on the GNSS reference clock. Notably, the DVB standard has been adopted as the reference specification for Telesat’s LEO (Low Earth Orbit) network [7], providing the proven Commercial On-The-Shelf (COTS) product in the market to satellite communication operators. Moreover, the potential to enable UEs for GEO satellite communication to support the coexistence with LEO satellites through software upgrades without new hardware is capable of supporting multi-orbit satellite constellations. Reviewing the DVB-RCS2 standard was published in 2011 for the GEO satellite [8]. It has been developed to enhance spectral efficiencies compared with the previous DVB-RCS system to satisfy the high demand for broadband service in the Ka band. In particular, the utilization of distributed pilot symbols and postamble, as well as preamble in a transmission packet, can be of help to the timing and frequency offset estimation accuracy. As a result, it can lead to Packet Error Rate (PER) performance improvement when operating the demodulator under much lower SNR conditions. The physical layer of DVB-RCS2 specification has two different modulation schemes. One is linear modulation like Phase Shift Keying (PSK) and Quadrature Amplitude Modulation (QAM), and the other is nonlinear modulation like Continuous Phase Modulation (CPM). Both schemes were selected as the mandatory technologies in the UTs due to their unique strengths in various service applications from a technical perspective. Meanwhile, the related research has been extensive, but there has been a scarcity of previous work focusing on the complete demodulator design. Most of the works have focused on the Carrier Recovery (CR) part, and there were limited descriptions of the demodulator in terms of functional block and performance results [9,10,11,12,13,14]. Specifically, the references [9,10,13,14] have dealt with practical or robust carrier synchronization to minimize the PER performance gap versus ideal conditions that work perfectly with STR in the AWGN channel. The reference [11] studied only the CPM modulation scheme in the DVB-RCS2 standard. The reference [12] described a complete DVB-RCS demodulator architecture that contains STR function but was considered a conventional scheme because mobility condition is not harsh in GEO satellite. The reference [14] showed a complete DVB-RCS2 demodulator structure and PER performance, but the STR scheme has been applied to employ the existing one that selects the maximum value as the optimum timing epoch position among 32 cross-correlation values within a symbol period. However, this approach did not consider the rapid timing and frequency phase variation that is characterized by GEO satellite conditions. The reference [15] has dealt with the demodulator of the Spread Spectrum (SS) mode to support DVB-RCS2 mobility. Under mobility conditions, a robust timing synchronization scheme is required to mitigate the Doppler effect. However, an SS scheme based on direct sequence can extend the number of chips in a packet by Spreading Factor (SF) length. It can provide the effect of extending the number of chips capable of estimating a time and frequency error.

In general, the previous works did not address the description of Log-on Burst (LB) and Control Burst (CB) transmission for initial access or resource requests, respectively, which are characterized by short packet length and a large time and frequency offset condition. In this paper, we clarify the necessity of new technology and propose a hybrid STR structure to improve the PER performance for the initial access mode. Note that our proposed structure is verified for its robustness in terms of a link-level performance-oriented approach. The main motivation and contributions of our work can be summarized as follows.

We propose a robust STR scheme for a GW, which is not susceptible to a large Doppler offset caused by the LEO satellite. In the GEO satellite, most conventional schemes considered limited Doppler offset and timing uncertainty environment due to the motion of UTs. On the other hand, for LEO satellites, it is not simple to maintain network synchronization between satellite and ground UTs because the UTs are used to receive a variable Network Clock Reference (NCR) signal due to the severe Doppler offsets caused by satellite motion. Thus, UTs are required to continuously estimate the differential propagation delay based on the serving satellite’s position and velocity within a beam. Furthermore, our scheme needs to consider the scenarios when UT should support high-speed vehicles such as airplanes or when UT should have a clock oscillator with low accuracy for cost-efficiency purposes [15]. Therefore, our proposed scheme can serve as a temporary solution until novel Forward Link Signaling (FLS) information, such as DVB System Information (SI), is developed to address Doppler and timing uncertainties in the DVB-RCS2 standard work for supporting LEO satellite communication [16].In the case of the 3GPP NTN standard, UTs are capable to compute and pre-compensate for the delay and Doppler frequency offsets due to the LEO satellite velocity and position through ephemeris information from GW’s and UT’s positions through the GNSS receiver when triggering access to GW [17,18]. Accordingly, there is a need to update the DVB-RCS2 standard, like the 3GPP NTN standard, to accommodate LEO satellite communication [16]. Until the amendment in the DVB standard, the proposed scheme can be effective as a viable option through GW receiver implementation technology within the standard. Even in the 3GPP standard, robust time and frequency synchronization enhancement based on non-GNSS operation has been required and discussed as a candidate technology for release 19. If there are situations where NTN UT is located temporarily with improper GNSS coverage or disruptions due to jamming, it may be necessary to perform initial access without the help of GNSS operations as implementation technology. The proposed scheme in this paper can contribute to improving performance in the case of the provisional non-GNSS operation when there are a large of timing and frequency offsets in initial access from UTs to GW.

The remainder of this paper is organized as follows. In Section 2, we describe the transmission model and explain the DVB-RCS2 demodulator structure for linear modulation. In Section 3, we discuss the existing schemes and present the proposed approach. In Section 4, we provide the comparative results and performance analysis. Finally, Section 5 concludes our paper.

## 2. Transmission Model Description

### 2.1. Transmission Structure for DVB-RCS2 Linear Modulation and Channel Model

The burst format for linear modulation of the DVB-RCS2 standard is depicted in Figure 1. It can be constructed according to the preamble, pilot field, and postamble symbol length flexibly by FLS configuration information. In Figure 1, the Half Guard Time (HGT) that can be described as a specific period of time inserted between consecutive data transmissions to avoid the overlapping with packets due to timing jitter and delay is the allocated time to avoid inter-burst interference by timing uncertainty. Preamble symbols are typically used to detect the presence and the start position of a burst within the timeslot, and they are also used to estimate the channel error using pilot field and postamble. The user field represents the modulated symbols with QPSK, 8PSK, and 16QAM constellations. The presence and contents of the preamble, pilot field, and postamble are determined by FLS from a Gateway. If pilot fields are required, they should be periodically inserted, and one pilot field needs to be followed by user field symbols. The postamble is appended after the final user field symbol is placed in the burst. The set of burst design parameters is {nSymbol, PreambleLength, PostambleLength, UserFieldLength, PilotFieldLength}, and it is signaled by the Broadcasting Composite Table (BCT) of the FLS SI through a forward link [8], where the GW sends to UTs via satellite transponder. In Annex A of the RCS2 specification [8], the list of reference waveforms (essential set) is tabularized, and it is a mandatory requirement that UTs shall have the capability to produce them. As we can see in Figure 1, each timeslot consists of a single burst and HGT at both each start and edge of the burst. The GT should be minimized because it should not waste resources in the TDMA system. However, it should be at least necessary and be allocated when considering network synchronization timing errors, UT and satellite motion, beam footprint size, transmission power switch on/off transient time, and more. Therefore, the length of HGT in the symbol time unit is, in general, determined by system parameters such as symbol rate, clock timing jitter, and drift by Doppler and NCR, which is a vital component in network synchronization providing a common timebase from GW to synchronize with UTs and extra. For LEO satellite systems, the maximum Doppler shift and drift can be calculated from the carrier frequency, elevation angle, and altitude of the LEO satellite. For example, the maximum Doppler shift ±420 kHz and Doppler rate ±260 Hz/s and around ±20 μs timing offset per one second can occur in case of 20 GHz frequency and 1200 km altitude. Furthermore, the maximum differential delay between GW and UTs in a beam can be determined by LEO antenna beamwidth, satellite altitude, and UT’s position, and it amounts to 3.18 ms [18].

Figure 2 shows the transmission structure of each UT toward the return link. It supports the construction of configurable burst format of DVB-RCS2 physical layer and up-conversion in low Intermediate Frequency (IF) or L-band frequency band directly. As we can see in Figure 2, the composite chain of the channel model is not aligned with the reality in order. For example, the Solid State Power Amplifier (SSPA) block should be placed after the up-conversion block. For simplicity, it works at a lower operational clock rate. In addition, each UT has an independent burst Timing Offset of Arrival (ToA), which refers to the difference in the arrival times of signals when compared to a reference time in an HGT interval, sampling clock drift, and carrier offset with uniform distribution every transmission time. Furthermore, the Adjacent Channel Interference (ACI) by power unbalance between inter-carriers and aggregated phase noise models, such as a terminal, satellite transponder, and GW operational clock-based Phase Lock Loop (PLL), should be incorporated into the return link channel model, as in Figure 3.

### 2.2. Demodulator Structure for DVB-RCS2 Linear Modulation

Figure 4 illustrates a high-level functional block diagram of the TDMA burst demodulator for the DVB-RCS2 standard. In the next part, we briefly review the functionality of each block. The received signal is bandpass filtered, sampled, and digitally down-converted to baseband signal through an Analog Digital Convertor (ADC) block. The continuous time version of the received signal can be modeled as follows:(1)$$r(t)=\sum_{k}a_{k}p(t-\tau -kT)e^{\left[2\pi \Delta ft+\phi (t)+\phi_{o}\right]}+w(t),$$
where *T* is symbol duration, τ is a time delay, Δf is Carrier Frequency Offset (CFO) or Doppler shift offset, ϕ(t) is time-varying phase noise such as Wiener processor Doppler rate, and $\phi_{o}$ is arbitrary initial phase offset, respectively. $a_{k}$ can be a sequence of modulated transmission symbols that contain preamble, distributed pilot, and postamble symbols. p(t) is the composite of channel impulse response reflecting pulse shape, static channel, and Matched Filter. The variable *k* means symbol time index, and *w(t)* is a complex AWGN channel.

We assume that complex I/Q baseband samples are stored in digital memory, and a channelizer process like Figure 4 is performed. The serialized baseband signal from multicarrier signals passes through a Matched Filter (MF). The burst detector block consists of a Unique Word (UW) detector and coarse frequency recovery. UW detector simply is a correlation process using differential, non-coherent, and coherent schemes depending on an amount of CFO, which refers to the difference between the actual carrier frequency of a received signal and nominal carrier frequency of a transmitted signal due to Doppler or inaccuracy of clock generator. Coarse frequency recovery block roughly estimates CFO through the reuse of functional block in the process of differential UW detector and reduces the amount of a large CFO. In the UW detection process, coarse STR can be achieved so that the starting point of the burst in a timeslot is roughly determined. Strictly speaking, we can first identify the burst start position in a timeslot by symbol time index counter. Of course, if the correlator used for the burst detector may have multiple samples per symbol, the start position can be more accurate in a sample time index, but the receiver complexity should be increased. As the burst detector operates in a sample time without the information of an accurate sampling point with maximum Signal-to-Noise Ratio (SNR) in a symbol time duration, the CFO and the timing offset estimate can be less precise. Subsequently, the fine STR is to obtain the exact sampling time using the Oerder and Meyer (O&M) algorithm [19] with four samples per symbol, and fine frequency recovery is accomplished with the *FFT* or the Mengali and Morelli algorithm [20]. To elaborate further on the STR scheme, also known as symbol synchronization or clock recovery, is a critical process in digital communication systems that involves accurately determining the timing or phase of received signals.

Next, SNR and signal level are estimated in the level control block, and carrier phase recovery is operated by the second DA/Decision-Directed (DD) PLL algorithm or Colavolpe Barbieri Caire (CBC) algorithm [2,3]. For bit conversion from a symbol, the Log-Likelihood Ratio (LLR) for each bit from the constellation label is computed in a soft de-mapper block, and the erroneous information bits by the channel are corrected by a 16-state duo binary turbo decoder.

## 3. Robust Symbol Timing Synchronization

### 3.1. Previous Works

As already mentioned in Section 2.2, the STR function can be divided into two parts: coarse timing recovery and fine-timing recovery processing. Typically, the coarse recovery part is to find the start of the burst during the HGT of a timeslot, and the fine part is to find the exact sampling time during the symbol interval of a burst.

In most previous works and DVB-RCS2 demodulator designs, feedforward architecture-based NDA synchronization is prevalent for fine-timing recovery. In this regard, O&M introduced an NDA square law estimator using multiple samples per symbol, and others were suggested with a similar principle, i.e., different nonlinear functions such as Absolute Value Nonlinearity (AVN) arithmetic and fourth square, as in Figure 5 [20]. They exploit the presence in the spectrum of the squared output of the Matched Filter in a demodulator of a frequency component at *f* = 1/*T*. Then, they use the delay property of the Fourier Transform to relate the phase of the spectrum to the time delay as follows:(2)$$r(t-\tau )↔R(f)\cdot e^{-j2\pi \tau f}.$$

After the coarse STR completes the burst detection process, the fine STR using the O&M algorithm can be applied to the entire burst duration for timing offset estimation. Coarse timing and CFO frequency recovery can be performed by pre-compensation function on the UT side through FLS signaling information that contains the satellite position, velocity, and common timing offset of UTs in a beam. Even though coarse synchronization can be achieved by an accurate GNSS receiver, fine synchronization should be necessarily performed in the receiver.

The mathematical expression of the O&M algorithm can be expressed as follows:(3)$$\hat{\tau }=-\frac{T}{2\pi }\arg\left\{\sum_{k=0}^{NL-1}{\left|r\left(kT_{s}\right)\right|}^{2}e^{-j\frac{2\pi k}{N}}\right\},$$
where $\hat{\tau }$ is the estimated timing error, *T*_s_ is the sampling duration, *T* is the symbol duration, and *N* is the oversampling factor per symbol, which is equal to 4. The variable *L* is the number of symbols used for timing error estimation. The term *r*(*kT*_s_) is the input sample of the STR block. The computation is based on a large number of symbols to grow the reliability of estimation and to reduce the effects of noise. This estimator is very simple for implementation but not the best in the case of a short packet for log-on and control burst transmission mode under low SNR and a large frequency offset channel. To overcome this drawback, a DA-based Maximum Likelihood (ML) estimator can be considered. The details will be described in Section 4.2.

In our paper, we also derive the fine-timing processing step based on the ML estimator. As a measure of performance assessment, we can compute the standard deviation of the timing estimation error and compare it to the Modified Cramer–Rao Bound [21], given by
(4)$$E[{\left(\hat{\tau }-\tau \right)}^{2}]⩾\mathrm{MCRB}(\tau )=\frac{-1}{2g_{\alpha }^{\prime \prime }(0)}\times \frac{1}{L}\times \frac{1}{\frac{E_{s}}{N_{0}}},$$
where $\hat{\tau }$ is the timing estimation, and $g_{\alpha }(\cdot )$ is the raised cosine function with roll-off α. The right-hand side of Equation (4) will be derived in Appendix A.

### 3.2. The Proposed Structure

From Equation (1) in Section 2.2, the objective of timing offset synchronization is to determine the desired timing offset, i.e., sampling timing estimate $\hat{\tau }$ from the observation. If the sampling on time may be aligned with the symbol on time by the decimator, the log-likelihood function for the timing offset τ and carrier offset $\theta_{n}=2\pi \Delta fn+\phi (n)+\phi_{o}$ given the observed signal *r(t)*, which incorporates the known data symbols $\left\{a_{k}\right\}$ for *k* such that n−L<k≤n, denoted by vector $a_{k}$ during the timing offset estimation
(5)Λ(τ,θn)=2NoRe[e−jθn∑k=n−L+1nak*r(kT−τ)],
where $\theta_{n}$ is carrier offset at nT, which may be known or unknown during the detection. From the log-likelihood function Λ(⋅) of τ and $\theta_{n}$ of Equation (5) when carrier offset information is unknown and very low SNR condition.

The maximization of the likelihood function typically entails a two-step operation. The first step is called a coarse search to compute $\Lambda (\tau ,\theta_{n})$ over timeslot duration, and then the symbol time index that maximizes $\Lambda (\tau ,\theta_{n})$ is chosen during burst known symbol length *L*, where *n* = 0, 1, …, L−1. In practice, the peak search from the cross-correlation scheme between the received symbol and the known symbol is used as the maximization method. The second step is a fine search to find the maximum within a symbol duration using interpolation or oversampling in time domain computation. However, coherent, non-coherent, and differential correlation scheme in the time domain that is typically used as cross-correlation scheme is not robust to high Doppler and low SNR condition because it is difficult to collect signal energy due to rapid phase rotation and noisy component. Therefore, this paper proposes to find peak as a coarse search through the consecutive *FFT*, which is widely used to analyze the characteristic in the frequency domain from a sequence of discrete time signals and magnitude computation during a timeslot. The proposed scheme can be regarded as an approximate ML approach and is to design a finite discrete time and frequency bin in the coarse estimate step and try to fine search within a high hypothesis area.

We assume that the *P* (<*K*) known symbols composed of preamble, postamble, and distributed pilot symbols are given, and *K FFT* size is given. This approach requires PK computation for each sample time position. However, to perform the coherent correlation during the constant phase, the partial coherent correlation approach is introduced in the initial step. It is indicated as the subcorrelation window in Figure 6. The output complex signals, which are the received samples multiplied with the complex conjugate vector values of known symbols, are accumulated and indicated as the partial complex vector sum. According to partial coherent correlation length and *FFT* size, the computation complexity can be reduced drastically. The position n with the maximum magnitude of *FFT* output in each sample time can have the most probable CFO estimate value among discrete K bin CFO hypothesis values.

Based on the grid area to acquire a CFO estimate, the fine-timing search process can be performed among local maximum values at the same time. The *FFT* magnitude output $FFT\left\{\Lambda (\tau_{n})\right\}$ is shown in Figure 7, and the blue stick is represented as the maximum one in every sampling time index. The interpolation scheme can provide the fractional sampling time offset estimate $\hat{\tau }$ between n and (*n* + 1). Specifically, $FFT\left\{\Lambda (\tau_{n})\right\}$ is revealed as the maximum among all Λ(τ) in the first step. Fine-timing estimation to find the expected maximum position located in the transparent blue stick from Figure 7 as the second step, can be expressed in
(6)$$\hat{\tau }=\frac{F_{n-1}-F_{n+1}}{2F_{n}-F_{n-1}-F_{n+1}},$$
where $F_{n-1}=FFT\left\{\Lambda (\tau_{n-1})\right\}$, $F_{n}=FFT\left\{\Lambda (\tau_{n})\right\}$ and $F_{n+1}=FFT\left\{\Lambda (\tau_{n+1})\right\}$.

After the fine-timing estimator works, the sampling epoch can be adjusted by the parabolic interpolator in the time domain. As aforementioned, its complexity issue can be raised. However, it can improve the estimation range and accuracy together under a large Doppler offset even though the transmission packet size is short. As shown in Figure 6, the operational flow is as follows. The received sample signal is fed into the Rx sample buffer, and it is multiplied by complex conjugated preamble and postamble symbols on a multiple oversampling rate basis. Some partial correlation sum in complex components during the subcorrelation window should be a complex signal vector, and *FFT* computation is performed after zero padding according to the predetermined *FFT* size. After *FFT* computation, signal power is calculated with a complex vector, *I* + *jQ,* in the frequency domain. The outcome can be quite reliable because it is measured in a frequency domain that is not vulnerable to the CFO. Moreover, *FFT* computation is simplified because most of the samples in the time domain are filled with zero. After coarse STR by burst detector, the process is operated to find a peak of signal power during a limited timing offset interval in the frequency domain. The discretized peak value can be found through threshold crossing like Figure 6 or MAX strategy (i.e., find maximum value in a given duration), and the timing offset estimate $\hat{\tau }$ is refined by interpolating more accurate peak position (*n* + $\hat{\tau }$) through fractional oversampling time positions (*n*−1, *n*, *n* + 1) indexed from the maximum correlation peak and its adjacent peak value, as in Figure 6. At first glance, the *FFT*-based algorithm looks challenging in terms of H/W implementation because the processing latency of *FFT* computation is quite large. Therefore, it can have time constraints in Field Programmable Gate Array (FPGA) due to consecutive operation during a limited range by residual timing offset after coarse STR. However, since its processing interval is not significantly large, it can be solved through the appropriate pipeline design to relax timing constraints. The DVB-RCS2 standard operates in Frequency Division Duplexing (FDD) mode, similar to conventional satellite communication. However, if the proposed scheme is required to operate in Time Division Duplexing (TDD) mode, as in terrestrial networks, it is essential to consider that the complexity may increase due to the need for parallelization of operations. This parallelization is necessitated by the processing time constraints imposed by the demodulator.

## 4. Numerical Results

### 4.1. Simulation Condition

When a LEO channel is considered, the additional guard time should be secured for initial access and synchronization maintenance. Thus, we assume that the maximum HGT is rather large. In addition, for higher symbol rate support by 10 Mbaud (i.e., Mega symbol per second), the increase in HGT is inevitable in terms of symbol length. On the contrary, the Doppler can be larger when taking into account low symbol rate support by 64 Kbaud (i.e., Kilo symbol per second) and 128 Kbaud. In [12], the HGT length is set to 14 symbols in a Traffic Burst (TB). Here, we take into account all worst cases summarized in Table 1 according to LEO channel characteristics as illustrated in Section 2.1. For LB bursts in the RCS2 system, they are set to a maximum of +/−200 symbols that correspond to 20 μs. For simplicity, we assume that nonlinearity components such as SSPA, Adjacent Channel Interference (ACI), and phase noise condition are negligible because they are not explicitly related to fine STR operation in terms of functionality. From a system point of view, nonlinearity can significantly influence the system’s performance unless its requirements are defined. This paper includes the consideration that the received signal can be impaired by nonlinear channel implicitly because the channel model is built in low SNR and high CFO conditions. Consequently, the received signal distortions by inter/intra channel interferences incur an extra SNR loss in the demodulator. Therefore, we expect that the STR performance results related to the proposed scheme remain consistent under such an environment. This paper includes the consideration that the received signal can be impaired by nonlinear channel implicitly because the channel model is built in low SNR and high CFO conditions. Consequently, the received signal distortions by inter/intra channel interferences incur an extra SNR loss in the demodulator. Therefore, we expect that the STR performance results related to the proposed scheme remain consistent in such an environment. As addressed in Section 2.1, each burst has a frequency and timing offset selected from a uniform distribution over the entire uncertainty range by maximum.

In the case of waveform ID, we selected representative formats, as summarized in Table 2. For the other Modulation and Code rate (MODCOD) pair related to waveform ID, it is expected that the O&M algorithm without using known (i.e., preamble, pilot, and postamble) symbols can be better than the proposed one intuitively because the ratio of known symbols length in a burst can be reduced if the demodulator can work at higher SNR condition. Consequentially, the estimation accuracy of the proposed one is deteriorated due to the limitation of available known symbols.

### 4.2. PER Performance Assessment

Firstly, we compare the PER performances of the O&M algorithm of the conventional scheme and the proposed one for different waveform IDs [8,22]. The decoding algorithm to be used for the turbo decoder is the log-MAP (Maximum A Posteriori) algorithm. The iteration number of the turbo decoder is set to 8. In the legend of the chart in Figure 8, the symbolic marks represent the applied algorithm scheme, HGT, and CFO in order, respectively. For example, O&M-H100-F2 means that the O&M algorithm is used, HGT has a maximum 100 symbol length, and the normalized CFO value is 2% vs. symbol rate. Here, a blue curve with a diamond mark stands for ideal PER performance in AWGN and perfect synchronization conditions which can be found in chapter 10.2 of the reference [22]. Note that in Figure 8, we are mainly interested in the two specific PER points of 10^−3^ or 10^−5^, where they are related to the achievable Quasi Error Free (QEF) in the physical layer.

As we can see in Figure 8, the *FFT* algorithm we proposed always shows better performance than the O&M algorithm in terms of PER performance for LB, CB burst, and Short Traffic Burst (STB) 3 (i.e., waveform ID#3) burst regardless of channel condition, as in Figure 8a–c. Meanwhile, the O&M algorithm is superior to the *FFT* scheme in the Long Traffic Burst (LTB)3 (i.e., waveform ID #13) waveform, as in Figure 8d. Intuitively, we analyze that the performance can be determined by the known symbol length or the ratio of known symbols in burst for the DA estimator (*FFT*) and by the estimation length for the NDA estimator (i.e., O&M scheme). The SNR performance gap in Figure 8b between the two schemes is remarkably large, and this can be interpreted as the CB burst has the shortest length among waveform IDs and has relatively many known symbols compared to other waveforms at the same time as in Table 2. As a result, we can draw the analytic findings from the simulation results. In the process of uplink initial access, the packet for control data transmission is typically characterized as short length but lengthy known symbols for reliable packet detection. In such a case, the proposed scheme can reduce the performance gap around SNR 1.5 dB at PER 10^−5^ from the ideal AWGN condition compared to the existing schemes.

### 4.3. Complexity and Performance Impact

In this subsection, we identify the impact of PER performance according to different *FFT* sizes when considering the receiver complexity and estimate accuracy. When we consider the scenario that the burst detector in Figure 4 can estimate coarse CFO and ToA and compensate them initially, the average amount of residual CFO and ToA becomes around 2% vs. symbol rate and 100 symbols when CB sends. In such a case, the impact of performance by the CFO seems to be negligible even though a smaller *FFT* size is applied. As we can see in Figure 9, the PER performance is not influenced by different *FFT* sizes for the proposed scheme. In addition, the proposed scheme can be effective in terms of operational clock rate because it can work at a symbol rate.

In Figure 10, we compare different estimators in terms of the normalized mean timing estimator error variance through computer simulation. The simulation condition for CB transmission was applied in the same manner as in Table 1 and Table 2, respectively. It shows that all timing estimator schemes are rather far away from the ideal bound in Equation (4) because of the residual CFO condition. Despite the result, it appears that the *FFT* scheme is the closest to the Modified Cramer–Rao Lower Bound (CRLB), as in Figure 10. The O&M that is used in Section 3.1 as the representative scheme and the AVN scheme show similar performance in the SNR range of interest from the presented PER charts. Overall, it can be shown that the accuracy measurements of STR estimators in Figure 10 are well aligned with the PER performance result in Figure 8b.

## 5. Conclusions

The robust fine STR scheme using known symbols is proposed for short burst packets such as LB and CB bursts for initial access. It can utilize *FFT* computation because signal power is not influenced by CFO in the frequency domain. The PER performance of the proposed scheme has been compared with the representative scheme among the existing schemes. The simulation results confirm the feasibility of the accurate timing estimation under the LEO satellite channel. We anticipate that the proposed scheme can be effective in short packets when UTs corresponding to VSAT try to access the network, initially in the presence of a large Doppler offset and timing uncertainty when proper GNSS operation is not available. In our future work, we will analyze the performance based on the proposed STR by adding the burst detector in the frequency domain. It can provide insights into packet arrival in random access, along with packet error induced by channel distortion. From a system perspective, this study will shed light on latency time that arises during the initial access procedure in the LEO satellite channel.

## Figures and Tables

**Figure 1 sensors-23-08320-f001:**
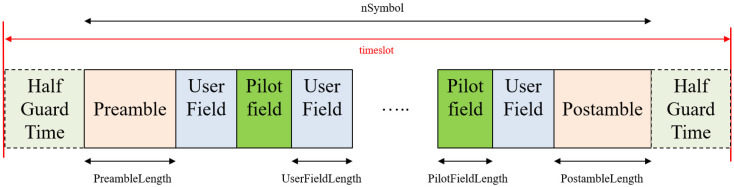
Burst structure in the timeslot for DVB-RCS2 linear modulation.

**Figure 2 sensors-23-08320-f002:**
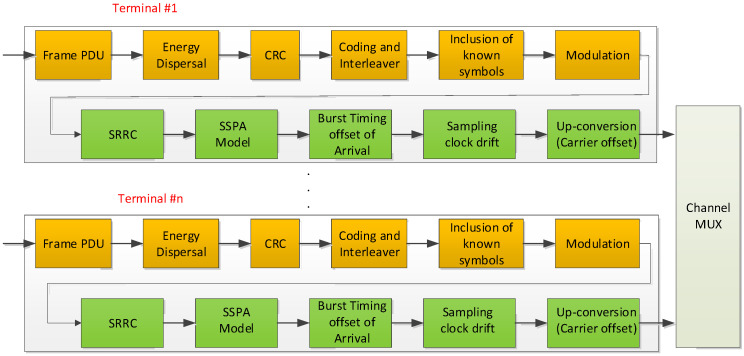
Transmission structure for DVB-RCS2 linear modulation at the Gateway point of view.

**Figure 3 sensors-23-08320-f003:**
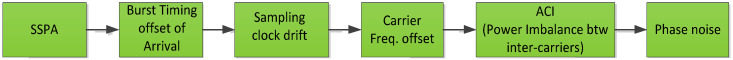
Channel model for return link transmission.

**Figure 4 sensors-23-08320-f004:**
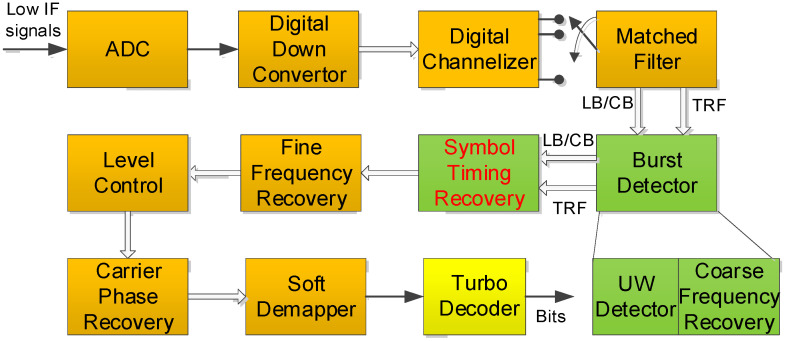
Demodulator structure of linear modulation for DVB-RCS2 standard.

**Figure 5 sensors-23-08320-f005:**
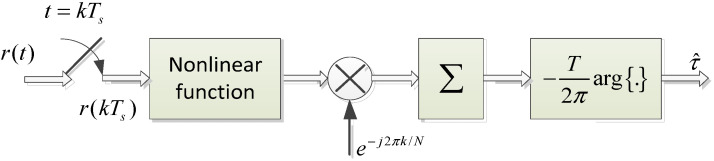
O&M algorithm architecture.

**Figure 6 sensors-23-08320-f006:**
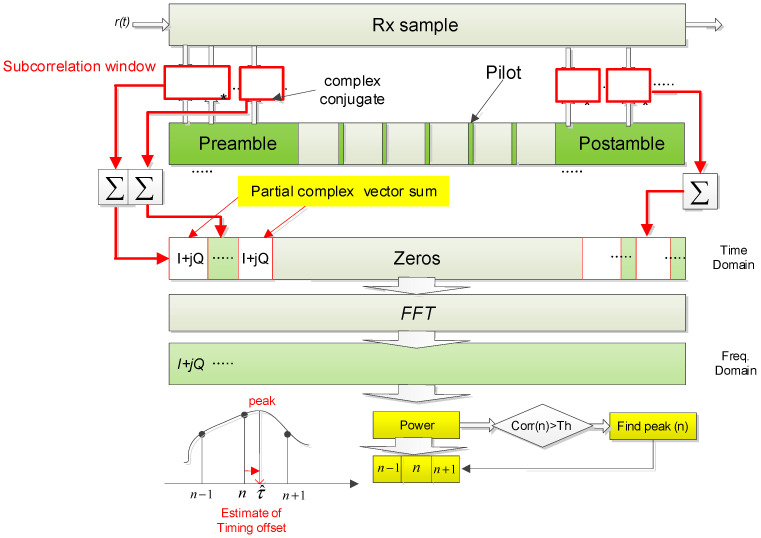
The proposed coarse and fine symbol timing estimator process for the initial access network.

**Figure 7 sensors-23-08320-f007:**
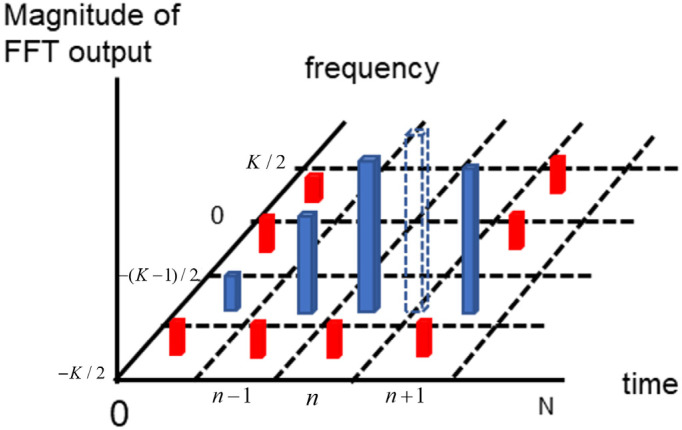
Magnitude of *FFT* output in discrete time and frequency grid.

**Figure 8 sensors-23-08320-f008:**
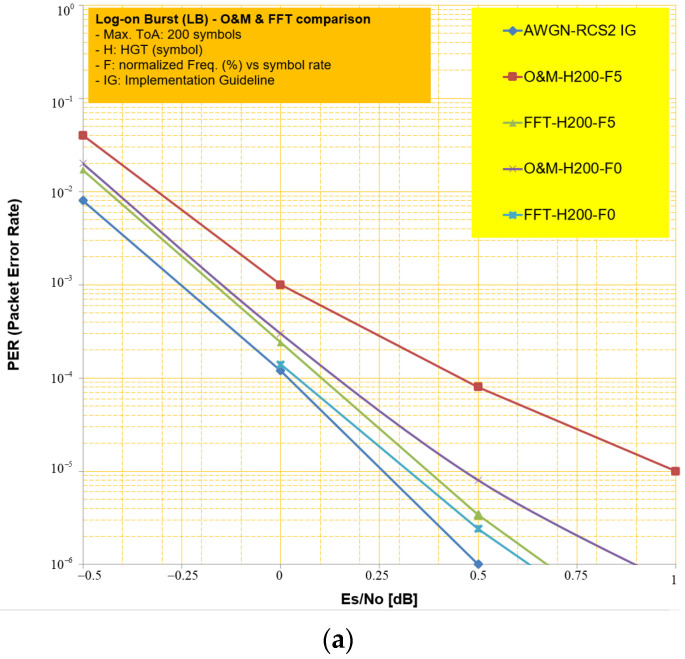
Comparative PER performance. (**a**) Log-on Burst (waveform ID#1). (**b**) Control burst (waveform ID#2). (**c**) STB3 (waveform ID#3). (**d**) LTB3 (waveform ID#13).

**Figure 9 sensors-23-08320-f009:**
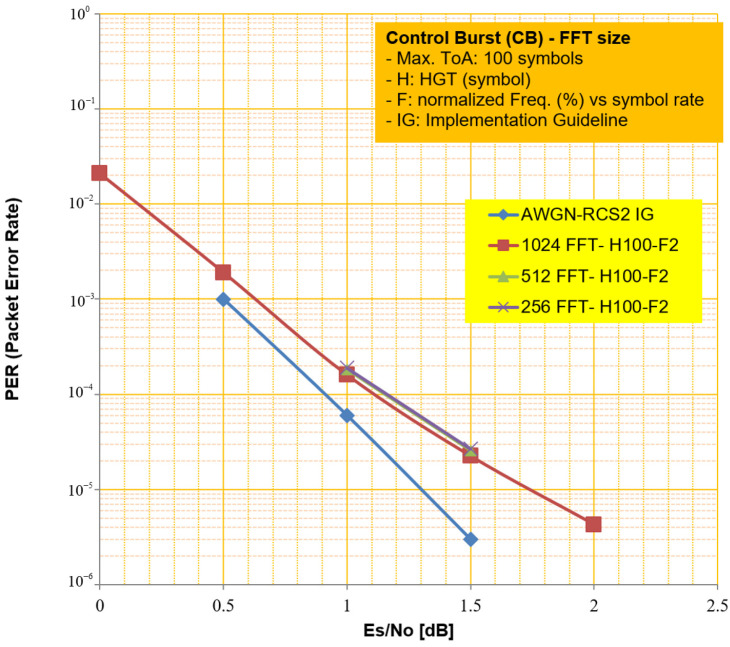
Comparative PER performance for different *FFT* sizes.

**Figure 10 sensors-23-08320-f010:**
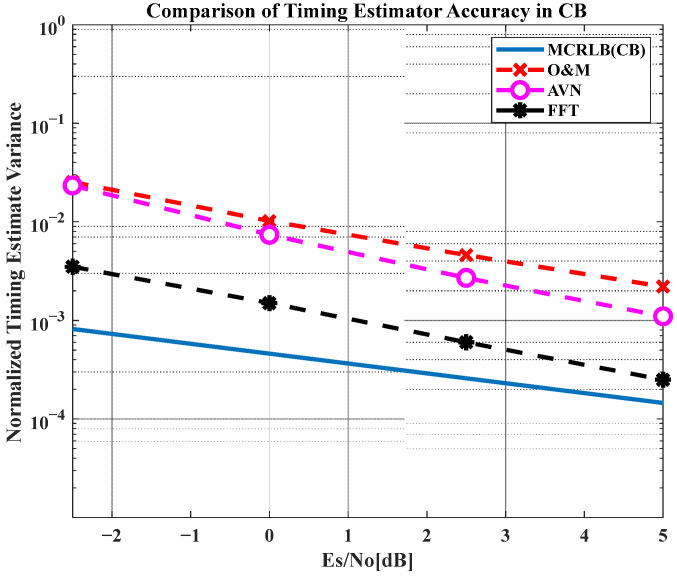
Normalized timing estimator accuracy in CB.

**Table 1 sensors-23-08320-t001:** Channel condition for simulation in symbol rate 10Mbaud.

Waveform ID	LB (ID#1)	CB (ID#2)	TB (ID#3, #13)
Maximum HGT (ToA in symbols) Uniform distribution	±200 symbols (±20 μs in 10 Mbaud)	±100 symbols (±10 μs in 10 Mbaud)	±20 symbols (±2 μs in 10 Mbaud)
Normalized maximum CFO vs. symbol rate Uniform distribution	±0.05 (±500 kHz in 10 Mbaud)	±0.02 (±200 kHz in 10 Mbaud)	±0.004 (±40 kHz in 10 Mbaud)

**Table 2 sensors-23-08320-t002:** Waveform ID and transmission parameters for simulation.

Waveform ID	Burst Length (Symbol)	Modulation	Code Rate	Known Symbol Length	The Ratio of Known Symbols in Burst
#1(LB)	664	QPSK	1/3	208	31%
#2(CB)	262	QPSK	1/3	94	36%
#3(STB3)	536	QPSK	1/3	80	15%
#13(LTB3)	1616	QPSK	1/3	140	9%

## Data Availability

The data presented in this study are available on request from the corresponding author.

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
