# Peer review of "Robust Symbol Timing Synchronization for Initial Access under LEO Satellite Channel"

_sensors, 2023, doi:10.3390/s23198320_

Round 1
Reviewer 1 Report
Overall, the paper is well written and structured, and the proposed solution is well described and assessed using a sound methodology. The problem addressed in this paper, however, could have been described in more detail to help the reader understand the significance of the proposed solution. This could be done by extending the introduction to cover the importance of LEO satellites today and how the problem addressed in this paper affects the use of LEOS. Maybe mentioning some applications and services that rely on LEOS to function properly and how the problem tackled by the author affects these applications and services.
Author Response
"Please see the attachment."

Reviewer 2 Report
This paper proposed a robust symbol timing synchronization scheme for return link initial access based on Digital Video Broadcasting-Return Channel via Satellite 2nd generation (DVB-RCS2) system for Low Earth Orbit (LEO) Satellite Channel.
Generally, the authors seem have done a solid work, however, some parts are not well written and need to be further clarified. The reviewer has the following comments.
1. It is suggested to add brief background and challenges at the beginning of the abstract to make it more clear. Besides, the authors should introduce more about the main work and novelty, only the last sentence talked about it.
2. The full name and abbreviations of “Low Earth Orbit (LEO) satellite” appears twice in the abstract.
3. It is suggested to introduce the following recent works in satellite communications
[R1]-[R4] fields to highlight the state-of-art of this paper:
[R1] “Secrecy-energy efficient hybrid beamforming for satellite-terrestrial integrated networks,”
IEEE Transactions on Communications, vol. 69, no. 9, pp. 6345-6360, Sep. 2021.
[R2] “SLNR-based secure energy efficient beamforming in multibeam satellite systems,” IEEE
Transactions on Aerospace and Electronic Systems, vol. 59, no. 2, pp. 2085-2088, Apr. 2023.
[R3] “Supporting IoT with rate-splitting multiple access in satellite and aerial-integrated networks,” IEEE Internet of Things Journal, vol. 8, no. 14, pp. 11123-11134, Jul. 2021.
[R4] “Secure and energy efficient transmission for RSMA-based cognitive satellite-terrestrial networks,” IEEE Wireless Communications Letters, vol. 10, no. 2, pp. 251-255, Feb. 2021.
4. Both the motivations and contributions should be much improved. The authors should explain why they investigate this topic and its novelty and meaning to practical applications. Furthermore, the second point of contributions is actually not the novelty point.
5. In figure 8, some points on curves are covered by labels. Besides, why there are only two points for “O&M-H4-F0.4” and “AWGN-RCS2 IG”?
Minor editing of English language is needed.
Reviewer 3 Report
The paper presents a robust fine STR (Symbol Timing Recovery) mechanism designed for short burst packets in Low Earth Orbit (LEO) satellite channels. This approach capitalizes on known symbols and FFT (Fast Fourier Transform) computations. Through a comparison of Packet Error Rate (PER) performances of the proposed and conventional algorithms across different waveform IDs, the research highlights the effectiveness of the novel scheme in maintaining accurate timing, especially in situations characterized by significant Doppler shifts and timing ambiguities. Simulation outcomes affirm its relevance, most notably when Very Small Aperture Terminal (VSAT) User Terminals seek to connect to the network under these rigorous conditions.
Major comments:
1. The authors' decision to overlook certain nonlinearity components, such as SSPA, ACI, and phase noise conditions, due to their perceived lack of direct relevance to fine STR operations requires more comprehensive justification. The potential influence of these assumptions on the outcomes warrants evaluation.
2. The paper occasionally employs intricate jargon that might impede understanding for readers not well-acquainted with the domain.
3. There seem to be multiple terms and abbreviations (like STR, CFO, HGT, O&M, FFT, etc.) used throughout the paper. Ensuring consistent use and clear definitions of terms would aid comprehension.
4. Expanding the comparison base by incorporating an array of algorithms or methodologies, beyond just one existing algorithm, could enhance the paper's depth.
5. The conclusion primarily restates the findings. It would be useful to include potential applications, implications for the broader field, and possible directions for future research.
6. The paper doesn't provide a strong linkage between the previous works cited and the current research. Knowing how this research builds on or diverges from existing literature can be invaluable.
7. While results are mentioned, there is a potential lack of in-depth discussion on why certain results were obtained, especially unexpected ones. Such discussions can provide insights into the workings of the system under study.
8. The paper seems to heavily lean towards the proposed method's advantages. A more balanced approach, discussing both the strengths and potential pitfalls of the proposed method, could lend more credibility to the research.
While the technical terms and abbreviations are used appropriately, a consistent and clear explanation of these terms (especially when first introduced) would aid in better understanding for readers who might not be deeply familiar with the topic.
Author Response
"Please see the attachment.

Round 2
Reviewer 2 Report
The authors have well addressed all my concerns, no further comments.
Reviewer 3 Report
Thanks for addressing my comments.